



# Year-to-year correlation, record length, and overconfidence in wind resource assessment

Nicola Bodini[1,2], Julie K. Lundquist[1,3], Dino Zardi[2], and Mark Handschy[4,5]

[1]Department of Atmospheric and Oceanic Sciences, University of Colorado Boulder, Boulder, Colorado, United States
[2]Department of Civil, Environmental and Mechanical Engineering, University of Trento, Italy
[3]National Renewable Energy Laboratory, Golden, Colorado, United States
[4]Enduring Energy, LLC, Boulder, Colorado, United States
[5]Cooperative Institute for Research in the Environmental Sciences, University of Colorado Boulder, Boulder, Colorado, United States

*Correspondence to:* Mark Handschy (mark.handschy@colorado.edu)

**Abstract.** Wind resource assessments predict future production levels from historical data. To characterize how year-to-year variability in past wind speeds affects the certainty of future predictions, we analyze 62-year wind speed records of 60 weather stations in Canada, and compare the actual levels of each station's final 20 years to "predictions" made from previous periods of varying duration. We estimate both median (P50) and 10% quantile (P90) levels using historical means and standard deviations,

validating estimator performance on statistically-independent "control" sequences made by randomly permuting the 62 annual values of each station's record. Errors of estimates made from the control sequences always decline with record length; the central half of the stations' exceedances falls within ranges of 44–55% (P50) and 85–95% (P90) for 42-year estimates. For the actual chronological records, on the other hand, error is lowest when estimates were made from short records (4–5 years) and increases with length after 15 years; for 42-year estimates the corresponding ranges are 0–45% (P50) and 36–100% (P90).

The strong biases reflect a nearly nationwide downward trend in recorded wind speeds, but even a near-zero-trend subset of 30 stations exhibits interquartile ranges of 24–73% (P50) and 80–100% (P90), both twice as large as expected. These findings show that serial correlation in wind speeds can persist across decades, and, if ignored, results in substantial overconfidence in estimated resource levels.

## 1 Introduction

Wind power is becoming less expensive and nowadays represents a very attractive low-emissions choice for electricity production. Its economy depends on generation plants being sited where enough wind blows to make development worthwhile; "resource assessments" are intended to identify such sites. Assessments can be carried out in different ways (Landberg et al., 2003), but predictions about future wind "climate" are based on historical observations. To provide sufficient history without decades of delay while measurements are collected at a target site, so-called Measure Correlate Predict (MCP) techniques are

used to infer wind resource levels by correlation with a longer record available for a nearby reference site. A wide variety of functional relationships have been investigated; see reviews by Brower (2012) and by Carta et al. (2013).





Resource assessment inaccuracies and uncertainties arise not only from less than perfect target/reference wind-speed correlation, including missing data (Salmon and Taylor, 2014), but also from many instrumental and model uncertainties (Mortensen et al., 2013; Lackner, et al., 2008). Here, though, we focus on the "Predict" step in resource assessment. We sidestep all the instrumental and model factors and their associated uncertainties and analyze only the extent to which historical wind speeds
can be used to predict future wind speeds.

Given wind's natural variability, an important question is how long a history is needed to both adequately estimate its mean level and to characterize the range of expected year-to-year variation. Previous authors have been of two minds. Justus et al. (1979) and Corotis (1980) both concluded that a single year's measurements were sufficient to give an estimate of the long-term mean accurate to 10% with 90% confidence, and that correlation with longer records at nearby sites offered the potential
for only marginal improvement. Concern over possible trends and the magnitude of variability over longer horizons, though, has driven examination of multi-decadal winds, using both direct observations (Palutikof et al., 1985; Earl et al., 2012; Früh, 2013; Azorin et al., 2014; Watson et al., 2015), and reconstructions using numerical weather prediction (NWP) reanalysis data (Palutikof et al., 1992; Albers, 2004; Bett et al., 2013; Kirchner-Bossi et al., 2015), with the uncertainties of reanalysis data examined by Rose and Apt (2015, 2016). Considering seven sites in the British Isles, each with more than 55 years of records,
Palutikof et al. (1985) suggested, "it is vital to consider the longest available wind-speed records, in order to obtain the best possible estimate of the range of variability of the future wind regime." Analyzing a 125-year reconstruction of geostrophic winds (winds computed from surface atmospheric pressure measurements) in Germany, Albers (2004) found that the error in "predicting" the mean of a given 20-year period decreased as the length of the preceding period on which the estimate was based increased, until the length of the estimating period had increased to about 35 years. On this basis he recommended
that resource assessments should be based on a historical record of at least 30 years' length. Taking a middle ground, Brower (2012, p. 163) stated "the benefit of going beyond about 10–15 years of reference data is limited." The effects of correlation between measurements made at times more than a few hours apart are rarely considered in published analyses of resource-assessment methodologies, although hub-height wind speeds exhibit anomalies persisting for at least 12 months (Klink, 2002) that complicate regression analysis (Pryor and Ledolter, 2010).

Here, we focus on how statistics derived from historical time-series records are used to predict the future exceedance levels employed in resource assessments. We use 62 years of homogenized monthly wind speed records from 60 Canadian stations (Environment Canada, 2016), as described in Section 2. To reveal the consequences of any unwarranted assumptions of statistical independence, we analyze for each station both its actual record and a "control" record of statistically independent values produced by randomly permuting the station's chronological values. We quantify estimation inaccuracy both as *exceedance*
*errors* and as *energy errors,* using concepts also introduced in Section 2. As we describe in Section 3, comparing the errors in estimates made from the chronological and randomized records unambiguously reveals that interannual correlations are responsible for larger than expected estimation errors and also for the growth of these errors with record length. This result is consistent with a slow power-law decay of year-to-year correlation, or "long-term persistence," which we discuss further in Section 4, and suggest that estimators explicitly accounting for year-to-year correlation behavior would enable both more
accurate predictions and a correct assessment of confidence.





## 2   Data and methods

### 2.1   Canadian wind speed dataset

We base our analysis here on one of the longer observational datasets of instrumental wind speed records available, a 62-year (1953–2014) record of monthly wind speeds from 156 Canadian meteorological stations (Environment Canada, 2016).
The stations stretch east-to-west from Vancouver to Halifax and north-to-south from the U.S. border to the Arctic Circle.
These data, from the National Climate Data Archive of Environment Canada, have been carefully homogenized by Wan et al. (2010). The homogenization process adjusted all wind speeds recorded at nonstandard anemometer heights to 10 m height using a logarithmic profile and surface-roughness-length data. Further, mean shifts from changes in anemometer height, type, or location, whether recorded in station histories or detected by statistical testing as part of the homogenization process, were also adjusted by comparison of each station's observational wind-speed record with a geostrophic wind reconstruction from independent surface pressure data. Using the homogenized data, Wan et al. (2010) found statistically-significant downward trends in wind speed over the period over most of Canada except the Arctic and Maritime provinces, which exhibited an upward trend. St. Martin et al. (2015) used a related wind speed dataset to investigate spatial correlation and geographic diversity in the wind resource.

### 2.2   From wind speed to capacity factors

For more relevance to wind-turbine energy production we convert monthly Environment Canada wind speeds to modeled turbine capacity factor (monthly electric energy production divided by turbine capacity). Since we begin with monthly averages the conversion is necessarily crude; with our interests focused on correlation effects, though, it suffices that low wind speeds translate to low capacity factor and high wind speeds to high capacity factor, with a variability scale comparable to that of real wind plant. Since we judge resource-assessment accuracy by comparing the exceedance levels in the final 20 years of each site's data to estimates based on the previous years' data, systematic conversion errors will not affect accuracy so long as we treat data from the final 20 years and from the preceding years in the same way.

As shown by Lackner, et al. (2008), the errors introduced by deriving capacity factors from a Weibull or Rayleigh statistical model of wind-speed distribution instead of directly from a wind-speed time series are only a few percent. We suppose that a monthly-mean value $u$ in the Environment Canada records arises as the average of more-frequently-sampled wind speed values $u'$ drawn from a Weibull distribution having cumulative distribution function $1 - \exp[-(u'/c)^k]$ with shape factor $k = 2$ and scale factor $c = 2\pi^{-\frac{1}{2}}u$; we convert speeds $u'$ drawn from this distribution into instantaneous capacity factors $x'$ using a power curve loosely modeled on that of a 1.5-MW-class turbine:

$$x' = \frac{v_1 + b \cdot v_1{}^3}{[v_0{}^\beta + (v_1 + b \cdot v_1{}^3)^\beta]^{1/\beta}},\qquad (1)$$

where $v_1 \equiv (au' - 3.5\,\text{m/s})/(13.5\,\text{m/s} - 3.5\,\text{m/s})$ is a normalized instantaneous hub-height wind speed, and parameters $v_0 = 2.244$, $b = 3.72$, and $\beta = 5.21$ control the power-curve shape. For all but the two windiest stations we set $a = 1.35 = (80/10)^{(1/7)}$, to account for wind speeds being greater at 80-m hub height than at 10-m measurement height according to a wind-profile power



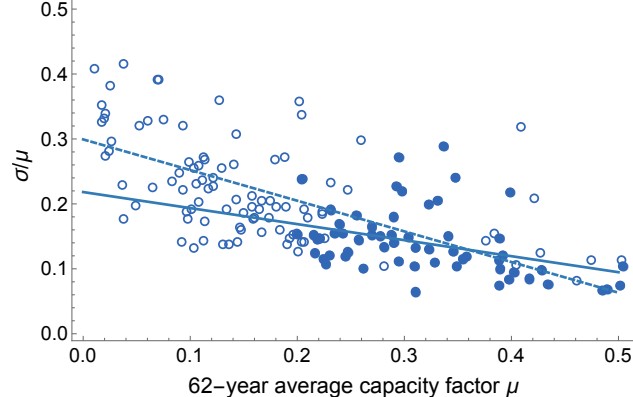

**Figure 1.** Ratio of the standard deviation $\sigma$ of the 62 annual average capacity factors to their mean $\mu$ versus $\mu$, for each station. Filled symbols: the 60 selected stations; open symbols: the discarded stations. Dashed line is the least-squares linear trend for all the 156 stations (slope of $-0.47$, $R^2 = 0.39$); continuous line is the least-squares linear trend for the 60 selected stations (slope of $-0.24$, $R^2 = 0.11$).

law with exponent appropriate for neutral stability conditions (Walter et al., 2009). For those two stations, with 10-m average wind speeds of 7.6 m/s and 8.3 m/s, we set $a = 1$ to avoid having sustained operation at rated power compress the modeled capacity-factor variability scale (or equivalently, 'to model a larger turbine'). We set $x'$ to zero for $au' < 3.5$ m/s. We do not model turbine cut-out as this complication has a negligible effect on variability scale. Averaging the $x'$ values for a given $u$ by

5  Monte Carlo simulation yields the corresponding monthly-mean capacity factor $x(u)$, which we parameterize as:

$$x(u) = 1.08 \frac{v_2^{2.5}}{1 + v_2^{3.27}}, \tag{2}$$

for $u \geq 0.922$ m/s (otherwise, 0) using a second normalized wind speed $v_2 \equiv (u - 0.922 \, \text{m/s})/(8.835 \, \text{m/s})$.

To avoid any effects of seasonal cycles in the subsequent analysis, we average the 12 monthly capacity factor values in a calendar year, and then work with these annual averages, 62 per station. To deal with data gaps, unavoidably present in such

10  an extended dataset, we assign null weights to missing monthly data. All the statistical functions calculated in the following analysis are thus weighted to take into account how many data points are available.

### 2.3  Stations selection process

In line with our focus on energy resource assessment, we eliminated from further consideration those stations with wind speeds too low for practical wind turbine deployment. We calculated, for each station, the ratio of the (weighted, always according to

15  the number of available data) standard deviation $\sigma$ of the 62 annual average capacity factors to their 62-year weighted average $\mu$. This simple indicator of variability is plotted vs. $\mu$ in Fig. 1. The least-squares linear trend in the plot reveals that the less windy sites tend to be more variable ($R^2 = 0.39$), in accordance with the findings of Rose and Apt (2015). For each of the 156 stations, we further calculated the final 20-years average capacity factor $\hat{M}$, where 20 years are considered in our work as the expected lifetime of a wind plant, yielding the distribution plotted in Fig. 2. As the histogram shows, there is a significant





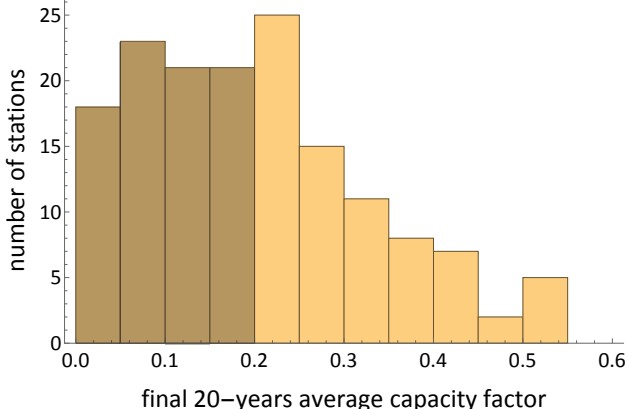

**Figure 2.** Histogram for the final 20-years average capacity factor $\hat{M}$ for the 156 stations. The 83 eliminated stations with capacity factor less than 20% are shaded.

number of stations that, using the 1.5 MW turbine, would have low capacity factors; to focus on the most plausible sites we eliminated from further analysis the 83 stations with capacity factor less than 20%. Imposing this cut-off also reduced the trend observed in Fig. 1 ($R^2 = 0.11$). We further eliminate another 13 stations that have more than a single isolated year with no or scant monthly data. Nine of the remaining 60 stations had a single year with no recorded monthly values; one had two years missing. In these cases, we fill in the missing annual value with the arithmetic mean of the values of the preceding and following years.

## 2.4 Trends and need for a 30-stations subset

Wan et al. (2010) found significant decreasing trends in wind speeds for most of the Canadian stations. We quantified these trends by calculating $\Delta\hat{\mu}$, the difference between the average capacity factors of the final 20 years and the next-to-final 20 years. Fig. 3 plots $\Delta\hat{\mu}$ versus $\mu$ for the 60 stations we analyze; 46 have a decreasing trend according to this definition. To enable our subsequent analysis to contemplate resource assessment statistics in the absence of such prominent trends we also contrive a smaller set of stations selected so that the average trend of the subset is near zero. This subset was selected by including all 14 stations with positive $\Delta\hat{\mu}$ and then adding stations with negative $\Delta\hat{\mu}$ until the sum of the set's $\Delta\hat{\mu}$ values was near zero, while attempting to keep the marginal distribution of $\mu$ values similar to that of the 14 positive-$\Delta\hat{\mu}$ stations and to end up with a distribution of $\Delta\hat{\mu}$ values roughly symmetric around zero. Filled dots in Fig. 3 show $\Delta\hat{\mu}$ versus the long term average capacity factor $\mu$ for the 30 stations in this subset.

## 2.5 Error categories and quantification

Resource assessments use quantiles or exceedance indices to characterize a site's wind resource. P50 expresses an energy production level chosen so that the fraction of time it is exceeded is expected to be half or 50%, and thus in some sense





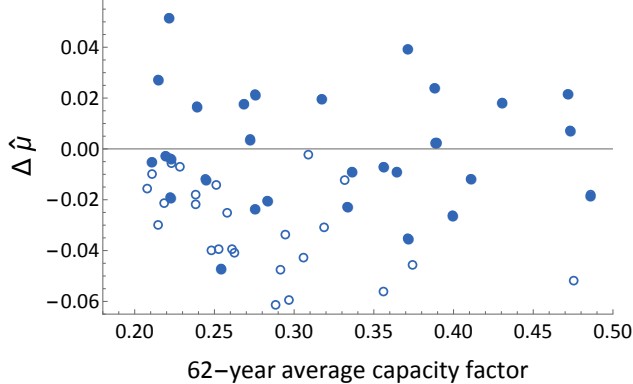

**Figure 3.** Difference $\Delta\hat{\mu}$ between the final 20 years average capacity factor and next-to-final 20 years average capacity factor versus 62-year long term average capacity factor $\mu$ for the 30 stations of the near-zero trend subset (filled) and for all the 60 considered stations (filled + open).

characterizes the expected "average" production level. A higher-denominated index value, such as P90, expresses a lower energy production level or "floor" that should be exceeded more often. Thus, it gives a sense of the financial "downside" risk. Estimators of this type are known in statistical quality control as $\beta$-expectation tolerance intervals (in contrast to $\beta$-content tolerance intervals) (Krishnamoorthy and Mathew, 2009). Although there exist many such estimators that do not rely

on detailed assumptions about the form of distribution whose quantiles are being estimated, knowing the form does enable more efficient estimation. Our annual-average capacity factor data often have nearly normal distributions, but, as we establish here, are far from independent year to year. Nevertheless, we proceed for illustration's sake under the simplifying assumptions that each station's annual capacity factor values $x$ are independent and identically distributed (iid) normal random variables, and derive P50 and P90 estimates from estimates $\hat{\mu}$ and $\hat{\sigma}$ of population mean $\mu$ and standard deviation $\sigma$.

We hold out the final 20 years of each station's record, representing a typical lifetime of a wind plant, as its "actual" production, and attempt to "predict" the actual production using estimates derived from capacity-factor values sampled from preceding years. To analyze the performance of the P50 and P90 estimates we count the (weighted) number of years in the station's final 20-year segment having capacity factor in excess of the estimator's value and divide by 20 (adjusted for weight), "expecting" the result to match the estimator's denomination (i.e. 50% or 90%).

To facilitate comparison to standard resource assessment metrics, we portray the error of our estimates or forecasts both as exceedance errors and as energy errors. Exceedance errors represent the difference of the actual from the targeted exceedance (e.g. "only 40% of the stations exceeded their estimated P50 level"). Energy errors represent the difference of the actual from the targeted capacity-factor quantile (e.g. "the median capacity factor of the station's final 20 years was 0.32 compared to an estimated P50 of 0.37"). Further, for both portrayals, we quantify both the spread and the "bias"—bias in the forecasting sense

of the difference between mean forecast and mean observation rather than in the statistical sense of difference between true value and expected value of an estimator.





Assuming each station's final 20 capacity-factor values are independent and symmetrically distributed, the distribution of energy errors should, as $j$ becomes large enough that the error of the estimated level becomes small compared to $\sigma$, tend towards a binomial distribution with $n = 20$, and $p = 0.5$ (P50) or $p = 0.9$ (P90). The quartiles of these binomial distributions are 0.4–0.5–0.6 and 0.85–0.90–0.95, respectively. With regard to the distribution of P50 energy errors, we define a statistic $t$

as:

$$t \equiv (\hat{M} - \hat{\mu}_j) \left[ \frac{(20-1)\hat{S}^2 + (j-1)\hat{\sigma}_j^2}{20 + j - 2} \left( \frac{1}{20} + \frac{1}{j} \right) \right]^{-1/2}, \tag{3}$$

where $\hat{M}$ is the mean and $\hat{S}^2$ the variance of the final 20 capacity factors. Under the assumption that both the final 20 and the preceding $j$ capacity-factor values are iid normal, this statistic has Student's $t$-distribution with $20 + j - 2$ degrees of freedom, and with mean (and median) of zero.

## 3   Results

### 3.1   Interannual variability

We first examine the potential effect of record length on resource assessment statistics by quantifying the interannual variability (IAV) of wind speed. For each station, we calculate the sample means $\hat{\mu}$ and variances $\hat{\sigma}^2$ for all possible contiguous segments of length $j$, allowing overlap: 62 segments of 1-year duration, 61 segments of 2-years duration, and so on, up to only one

62-year time segment, with weighted sample variance defined as:

$$\hat{\sigma}^2 = \sum_{i=1}^{j} w_i (u_i - \hat{\mu})^2 \cdot \frac{j_{\text{eff}}}{j_{\text{eff}} - 1}, \tag{4}$$

where the $w_i$ are the normalized data-availability weights of the wind-speed data $u_i$, and $j$ indicates the sample size (length in years of the data segment over which IAV is calculated). The second factor above, with $1/j_{\text{eff}} \equiv \sum_{i=1}^{j} (w_i)^2$, corrects for sample-size bias in a manner similar to the usual $N - 1$ term in the denominator of unweighted sample variance. As usual, we

define IAV as the coefficient of variation (the ratio of sample standard deviation to sample mean), but correct for bias arising from the square-root operation in $\hat{\sigma}$ as an estimator of the population standard deviation:

$$\text{CV}_j \equiv \frac{\hat{\sigma}/b(j_{\text{eff}})}{\hat{\mu}};$$

$$b(s) \equiv \left[ \frac{2}{s-1} \right]^{1/2} \frac{\Gamma(\frac{s}{2})}{\Gamma(\frac{s-1}{2})}; \quad \Gamma(s) = (s-1)! \tag{5}$$

Dividing $\hat{\sigma}$ by $b(j_{\text{eff}})$ gives, at least for normally distributed $u$, an unbiased estimate of the standard deviation (Kenney, 1940, p. 135). Then for each station we calculated $\langle \text{CV}_j \rangle$ as the average of $\text{CV}_j$ over all segments of length $j$. In this way, we get

a set of $60 \times 62$ coefficients of variation $\langle \text{CV}_{j,\ell} \rangle$, where $j$ indicates the length of the time segment ($j = 3, \ldots, 62$) while $\ell$ indexes the station ($\ell = 1, \ldots, 60$). Using a general technique that we will use repeatedly to explore the effect of year-to-year




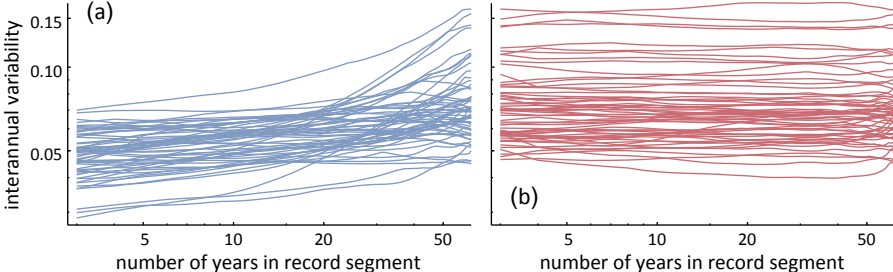

**Figure 4.** Average interannual variability $\langle \mathrm{CV}_j \rangle$ of wind speed versus time-segment length $j$ for each of 60 stations: (a) actual data, (b) randomly permuted data.

correlation, we randomly permuted the order of each station's 62 annual average wind-speed values, destroying any correlation, and then repeated the calculation of $\langle \mathrm{CV}_{j,\ell} \rangle$ using this "control" dataset. Figure 4 plots, with one trace per station, the average interannual variability vs. segment length for wind speed $u$.

Figure 4 compares the CV based on actual data to the CV based on randomized data, and clearly shows that, for the
actual chronological data (a), $\langle \mathrm{CV}_j \rangle$ systematically increases with segment length $j$, while, for the randomized data (b), it is independent of segment length. We also analyzed raw hourly data from Environment Canada that had not been homogenized and saw similar behavior, but with a faster growth of IAV with $j$. The increase in interannual variability with the length of the considered record can only result from year-to-year correlation. As our results show, *sample IAV is still increasing at the 62-year limit of our record,* which is already substantially longer than the datasets typically used for energy resource assessment.

**3.2  P50 estimation**

Under the assumptions of independence and normality, a "prediction" of the median, or P50, can be calculated from a sample of historical values simply as the sample mean $\hat{\mu}$. To evaluate how P50 estimation performance depends on record length, we calculated, for each selected station ($\ell = 1, \ldots, 60$ or, for the low-trend subset, $\ell = 1, \ldots, 30$), the weighted mean $\hat{\mu}_{j,\ell}$ of the immediately preceding $j$ annual capacity factors ($j = 1, \ldots, 42$), and then counted the (weighted) number of the final 20 years
having a capacity factor exceeding $\hat{\mu}_{j,\ell}$. Fig. 5 shows results for all 60 stations. For estimates made from the randomly-shuffled capacity-factor data (b), the fraction of final-segment values exceeding the estimate averages around 50%, and as $j$ increases more than half of the station trajectories in fact fall within the expected binomial interquartile range of 0.4–0.6. For estimates made from the actual data (a), on the other hand, the stations' exceedance fractions remain widely spread for all $j$ values, with much less apparent convergence towards any central value, with only 10 of the 60 stations having exceedance fraction within
the expected interquartile range.

Focusing on the 30-station low-trend subset, the histograms of Fig. 6 show that the distribution of exceedances for the chronological data (a) is very different from binomial, while the distribution of exceedances of an instance of random data (b) is very close to binomial. The differences from binomial distribution are statistically significant: Pearson's $\chi^2$ test $p$-values of magnitude of $10^{-10}$ (chronological data) vs. 0.7 (randomized data). The quartiles of the 30-station exceedances are 39–50–





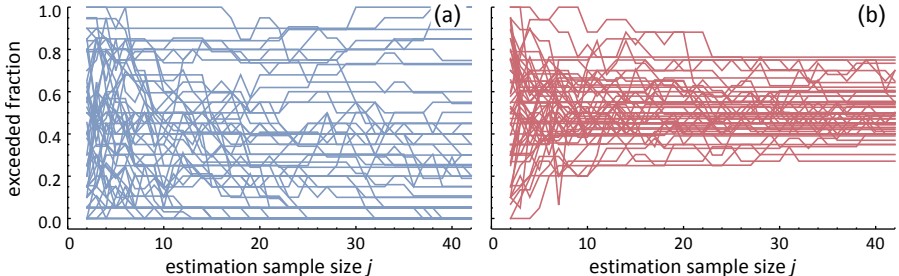

**Figure 5.** Fraction of each of the selected 60 station's final 20 years with capacity factor exceeding its P50 estimate $\hat{\mu}_{j,\ell}$ for actual (a) and randomized (b) data.

56% for the randomized data but 24–43–73% for the chronological data (the exceedances take values different from multiples of 1/20 because of our data weighting). Trends or no, year-to-year correlations increase the exceedance error range, here by a factor of nearly 2.5 compared to the expected 20% binomial interquartile range.

To characterize bias in the P50 estimator, we also calculate the exceedance fraction averaged across all 60 stations, as shown in Fig. 6(c). The curves show the weighted fraction of the $20 \times 60$ values that exceed their respective P50 estimate, equivalent to the average across the traces in Fig. 5 at each $j$ value. This perhaps comes as close as we can come to measuring what we desired to predict: the *expected* fraction of capacity factor values exceeding the estimator value. For the randomized data (red) the P50 estimator is indeed exceeded about half the time. However, for the actual data (blue) the exceedance fraction never rises above 45%, peaking for a $j = 4$ year sample, and declines further thereafter as record length is increased out to 42 years. This behavior reflects the widespread decreasing trend in wind speeds reported by Wan et al. (2010). When the analysis is repeated with the low-trend subset of 30 stations, Fig. 6(d), the average exceedances for both actual and random data vary around the expected value of 50% without the large excursion seen for the whole 60-station dataset. But, since the subset was selected to have just the property that average resource level over the final 20 years match that of the previous 20 years, this comes as no surprise. The estimator's bias for the larger set of 60 stations may be more indicative of expected behavior for instances where continuing trends cannot be separated *a priori* from random behavior.

To characterize the magnitude of the energy estimation error, we calculate the $t$-statistic according to Eq. (3), with $j = 42$ results shown for the low-trend set of 30 stations in Fig. 6(e) and (f). The 5–95% range (light horizontal bars) nearly matches the theoretically expected 90% confidence interval (black bar) for the randomized data (f), but is more than three times wider than expected for the actual chronological data (e). In the case of a hypothetical set of sites with an average capacity factor of 0.35 and 6% IAV, this increased error would translate into an interval of $0.27 < P50 < 0.43$ being required to capture 90% of the sites vs. the range $0.32 < P50 < 0.38$ expected for statistically independent annual capacity factors.

We also calculate the mean absolute energy error (MAE) of the P50 estimates:

$$\text{MAE}_j = \frac{1}{n} \sum_{\ell=1}^{n} |e_{j,\ell}|, \tag{6}$$



where the energy estimation error $e_{j,\ell}$ is defined as the difference between the median of the $\ell^{\text{th}}$ station's final 20 capacity-factor values and $\hat{\mu}_{j,\ell}$. Fig. 6(g) shows results for the whole set of 60 stations; Fig. 6(h) depicts the low-trend subset of 30. To distinguish the effects of year-to-year correlation we also compute $\text{MAE}_j$ using a randomly permuted instance of the 62 annual capacity factors for each one of the considered stations, shown as red traces in Fig. 6(g) and (h). The plots reveal that

for the randomized data the MAE always decreases as the sample size $j$ increases, as expected for uncorrelated data. However, for the actual data the result is quite different. Fig. 6(g) shows that, for the full 60-station dataset, the best prediction (lowest MAE) of the 20-year average energy production is reached using just a short 4–5 year segment immediately preceding the final target period, similar to the finding of Früh (2013). For the low-trend subset of 30 stations, the MAE for actual data again reaches a minimum at $j = 4$, but stays almost constant after that out to an estimation sample length of about 20 years.

It increases thereafter, although more slowly than for the data from all 60 stations. However, for both the set of 60 stations and the low-trend subset, the P50 estimation MAE for actual data always remains larger than the MAE obtained for randomly permuted data; interannual correlation increases error in estimating future energy production.

To provide additional insight into the effect of year-to-year correlation, we again estimate the P50 of each station's final 20-year segment, but now using a fixed 5-year sample, separated from the final segment by a $p$-year interlude. Fig. 7 shows

the P50 MAE for the 60 stations versus the length $p$ of the interlude, with the blue trace representing the actual chronological data and the red trace the randomly permuted data, as before. For the chronological data the 5-year time segment immediately preceding the target period produces the best P50 prediction while going further back in time consistently increases the energy error. For the randomized dataset, on the other hand, this effect is totally absent: the estimation error stays essentially constant regardless of how far back we look. A similar result is found also considering the low-trend 30 stations.

As these results show, year-to-year correlation has substantial impacts on the estimation of P50. When the preponderance of the stations appear to exhibit a secular trend it is perhaps not surprising that P50 estimates will exhibit bias. However, even for a group of stations without a predominant trend, where bias is essentially eliminated, error in the P50 estimate is still always more than would be expected on the basis of uncorrelated samples, for all record lengths. No improvement in error is gained by using records longer than 4–5 years, and in fact using records longer than 20 years actually degrades accuracy.

### 3.3  P90 estimation

The degree of resource variability, and hence financial risk, can be indicated by a "floor," or a production level enough lower than P50 that it is only rarely *not* exceeded. P90, for example, indicates a production level expected to be exceeded 90% of the time, or in 9 years out of 10. To estimate P90 we again proceed, for illustration's sake, under the simplifying assumptions that each station's annual capacity factor values $x$ are independent and normally distributed, enabling our P90 estimator $\widehat{q_{10}}$ to be

determined from the two parameters that completely characterize that distribution:

$$\widehat{q_{10}} = \hat{\mu} - k\hat{\sigma}, \tag{7}$$

where $\hat{\mu}$ and $\hat{\sigma}$ are the mean and standard deviation of the sample of annual-average capacity-factor values from which the estimate is being made (Wilks, 1941; Krishnamoorthy and Mathew, 2009, p. 295).



According to the definition of P90 we desire an estimator such that $E\{\Pr[x > \hat{\mu} - k\hat{\sigma}]\} = 0.9$, where $E\{\cdot\}$ denotes expectation and $\Pr[\cdot]$ probability. If the estimate is being made from a sample comprising $j$ iid normally-distributed annual capacity-factor averages $x$, then $x$ and $\hat{\mu}$ both have expected mean $\mu$, and have variance $\sigma^2$ and $\sigma^2/j$, respectively. Thus, $(x - \hat{\mu})/(\sigma\sqrt{1 + 1/j})$ has the unit normal distribution and $(j-1)\hat{\sigma}^2/\sigma^2$ has the $\chi^2$ distribution with $j-1$ degrees of freedom.

Since $(x - \hat{\mu})/(\hat{\sigma}\sqrt{1 + 1/j})$ then has Student's $t$-distribution with $j-1$ degrees of freedom, the constant $k$ needed for estimator $\widehat{q_{10}}$ in Eq. 7 is determined by:

$$k = Q_{t,j-1}(0.1)\sqrt{1 + 1/j}, \tag{8}$$

where $Q_{t,\nu}$ is the quantile function (inverse CDF) of Student's $t$-distribution with $\nu$ degrees of freedom. As shown in Fig. 8, these values of $k$ are larger than 1.282, the 10% quantile of the unit-normal distribution (Mortensen et al., 2013), especially for

small sample sizes, but approach it for large $j$, as expected.

To evaluate how P90 estimation errors depend on record length, we calculate P90 estimates using the weighted means and standard deviations of the immediately preceding $j$ annual capacity factors according to Eq. (7), and then count the (weighted) number of each station's final 20 years having a capacity factor exceeding the value of its P90 estimate, with results shown in Fig. 9. To distinguish the effects of year-to-year correlation, we again carried out this procedure for both the actual chronolog-

ical data and for an instance of each station's data having the 62 values randomly permuted. As can be seen in Fig. 9, which plots all 60 stations, while the exceedance values for the randomized data converge on 90% as sample size $j$ is increased, the values for the actual data remain widely scattered. The low-trend subset of 30 stations gives a similar result.

As for the P50 estimates, we examine the distribution of the fractions of the final 20 years' capacity factors exceeding the estimated P90 for the low-trend subset of 30 stations. If the final 20 years were independent, the exceedance counts would

have approximately binomial distributions, this time with $p = 0.9$ and interquartile range of 85–95%. Fig. 10(a) shows that the distribution of exceedances for the actual data is very different from binomial ($p$-value of Pearson's $\chi^2$ test of the order of $10^{-8}$), while the distribution for randomly-permuted data, Fig. 10(b), is quite close ($p$-value equal to 0.85). The quartiles of the 30-station exceedances are 85–90–95% for the randomized data but 80–95–100% for the chronological data: the interquartile range is twice the expected 10% range of the binomial distribution. Year-to-year correlation clearly shows its influence in

widening the confidence interval of a P90 prediction, even for a set of stations with little average long-term trend.

The average across all 60 stations of the exceedance of the final 20 capacity factors, as seen in Fig. 10(c), declines from 85% to 65% with increasing $j$ for the actual chronological data. For this dataset, estimator $\widehat{q_{10}}$ thus grossly underestimates risk when using long records, with up to 1 year in 3 falling below the supposed P90 instead of the desired 1 in 10. Suspicions about an improper definition of the estimator are allayed by its performance with randomized data, where the average exceedance in

fact stays about 90% essentially independent of $j$. Bias is largely eliminated when using the low-trend subset of 30 stations, as seen in Fig. 10(d). In this case, the average exceedance for estimates made from both the chronological data and the randomly permuted data vary around the expected value of 90%, without any large systematic trend.

Figures 10(e) and (f) show the distribution of energy errors for the low-trend subset. Energy error here is the difference between the 10% quantile of each station's final 20 years, calculated according to Definition 5 of Hyndman and Fan (1996),





and its P90 estimate made from the preceding $j = 42$ years. For the chronological data, half the stations have capacity-factor errors within the range $-0.031$ to $+0.014$, while for the randomized data the range is $-0.013$ to $+0.005$, or 2.4 times smaller.

We also calculate the mean absolute value of this error vs. $j$, with results for all 60 stations in Fig.10(g) and for the low-trend subset of 30 in Fig.10(h). Again, we reveal the effects of year-to-year correlation by also calculating MAE using a randomly permuted instance of each station's 62 annual capacity factors, with traces shown in red. For the randomized data, the MAE decreases as the sample size $j$ increases, as expected. However, for the actual chronological data, Fig.10 shows that for both all 60 stations (g) and the low-trend sub-set of 30 (h), the best predictions (lowest MAE) of the 20-year energy production is reached using a 10-year segment immediately preceding the final target period, and that records longer than 20 years *increase* the MAE. At $j = 42$, MAE for the low-trend subset reaches a value twice what would be obtained if the samples were independent. Moreover, as seen previously for the P50 estimation errors, for both the full 60-station set and the low-trend 30-station subset, the P90 MAE for actual data is always larger than the MAE obtained for randomly permuted data: year-to-year correlation increases P90 estimation error whether or not the data exhibit overall secular trends.

Our P90 estimator $\widehat{q_{10}}$ relies on separate estimates of sample mean $\hat{\mu}$ and sample standard deviation $\hat{\sigma}$. We perform a "thought experiment" to see which makes the larger contribution to P90 error. First, we calculate P90 estimates using the actual mean of the final 20 years instead of estimating it from a sample of preceding years, but still relying on the preceding years' sample estimate of $\hat{\sigma}$. Then, we turn the tables, and estimate P90 estimates using the actual standard deviation of the final 20-year segment, but with the the mean estimated from the preceding years. Of course an assessment engineer could never see the 20 target years, but the exercise can help us isolate the effects of errors in scale from the effects of error in location. Carrying out this thought experiment with estimators that still are expected to have 90% exceedance under our iid assumptions requires $k$-values different than those used before where both $\hat{\mu}$ and $\hat{\sigma}$ are estimated from preceding years.

For the first half of the thought experiment, we utilize the actual mean $\hat{M}$ of each station's final 20 annual capacity factors. In this case, the difference between a particular one of the final capacity factors $x_\alpha$ and the mean of all twenty can be written:

$$x_\alpha - \hat{M} = \frac{19}{20} \left( x_\alpha - \frac{1}{19} \sum_{i \neq \alpha}^{20} x_i \right) \implies$$
$$\sqrt{\frac{20}{19}} \left( \frac{x - \hat{M}}{\sigma} \right) \sim N(0,1). \tag{9}$$

For the sample standard deviation of the preceding $j$ years, $(j-1)\hat{\sigma}^2/\sigma^2$ has the $\chi^2$ distribution with $j-1$ degrees of freedom as before. Thus, the estimator for this special case can be calculated as:

$$\widehat{q_{10}}' = \hat{\mu} - \sqrt{\frac{20}{19}} \, Q_{t,j-1}(0.1)\hat{\sigma}, \tag{10}$$

where $Q_{t,\nu}$ is again the quantile function for the t-distribution with $\nu$ degrees of freedom. As can be seen in Fig. 11(a) and (b), bias for both chronological and randomly-permuted data is small in this case, regardless of trend: using the actual means of the final 20 years greatly reduces P90 bias.



For the second half of the thought experiment, we utilize the actual sample standard deviation $\hat{S}$ of each station's final 20 years, but estimate the mean $\hat{\mu}$ from the preceding $j$-years as we did originally. In this case the final twenty $x$ values and $\hat{S}$ are not independent, which complicates expressing $k$ in terms of standard probability distributions. Instead we approximate it by Monte Carlo, creating sets of samples of $j$ and of 20 iid normal-distributed random numbers (100,000 of each), and calculating $\hat{\mu}$ values from the $j$-member samples and $\hat{S}$ values from the 20-member samples. We then vary $k$ until approximately 1 800 000 of the 2 000 000 values in the 20-member samples exceed their sample's estimator. Bias for this P90 estimator behaves much like that of our original estimator: substantial bias for the larger set of stations and little for the low-trend subset, as seen in Fig. 11(c) and (d), respectively.

## 4   Discussion and Conclusions

The primary purpose of resource assessment is to quantify financial risk and returns, and to this end it is important that resource assessments quantify their degree of certainty. Using simple estimators generated from sample mean and variance ($\hat{\mu}$ for P50 and $\hat{\mu} - k\hat{\sigma}$ for P90), we have analyzed homogenized wind speed data from 60 weather stations in Canada, each with a 62-year record, with a goal of untangling the influence of interannual variability and year-to-year correlation on estimates of financial risk. By randomly permuting each station's record, we created control data sets with sequences guaranteed to be statistically independent. Checking the exceedance-level estimates P50 and P90 on these randomized sequences confirms several anticipated characteristics of the estimators. The frequency with which the estimated levels are exceeded for the randomized time series indeed averages around 50% and 90%, indicating that the distributions of annual-average resource levels are approximately normal, without much skew (which would bias sample mean $\hat{\mu}$ as an estimator of P50 or the resource median) or excess kurtosis (which would bias $\hat{\mu} - k\hat{\sigma}$ as an estimator P90 or the 10% quantile). Hitting the target exceedance frequency does require calculating $k$ for the P90 estimator from the $t$-distribution. Improperly setting $k = 1.282$, the 10% quantile of the normal distribution, would give P88 estimates when applied to samples of length $j = 10$, for example. With these P50 and P90 estimators, the mean absolute energy error (MAE) of the estimated capacity-factor quantiles falls monotonically with record length, as expected. We checked estimator performance by counting the number of years in the final 20 of each station's record having a resource level exceeding an estimate made from a preceding segment of the record. When the estimation segment was substantially longer than 20 years the distribution of the count was essentially binomial with $n = 20$, and $p = 0.5$ or 0.9. These results show that, save for the assumption of statistical independence of the input data, the estimation methods are adequately formulated.

The performance of the estimators using the actual chronological records is quite another story. When considering the entire set of 60 stations, both the P50 and P90 estimates exhibited strong bias, grossly over-predicting resource levels actually attained in the final 20 years of each station's record. This bias is consistent with widespread decreasing wind speeds identified by Wan et al. (2010). Errors only fell with record length for the first few preceding years (4 years for P50 and 10 years for P90); for records longer than 5 years (P50) or 18 years (P90), error rose with sample length. Using a a sub-set of 30 stations contrived to have near-zero average resource trend (same 30-station-average level for the final 20 years as for the preceding 20 years)





essentially eliminates estimation bias, but both energy and exceedance error spreads increase with sample length for records longer than 18 years, and are 2–3 times larger than either theoretical expectations or errors obtained from the same estimation procedures applied to randomized data. Thus, even absent overall trends, year-to-year correlation in the chronological data reduces P50 and P90 certainty.

The higher errors of the estimates made from the chronological data must arise from non-zero correlation (lack of statistical independence) since these data are identical to the randomized control data except for sequence. One might hope to account for the higher errors in estimates made from the correlated data in terms of an "effective number" of independent samples (Bayley and Hammersley, 1948; Corotis et al., 1977). For example, if the autocorrelation function of a sequence of wind speed measurements was found have an exponential form and to fall to $1/e$ in a lag time of $\tau = 6$ hours, then a sequence of hourly

measurements having total duration $T \gg \tau$ could be considered to comprise $j = T/(2\tau) = T/(12\,\mathrm{h})$ independent samples, and confidence intervals for its mean value could be correctly estimated using this value of $j$. However, this approach relies on the autocorrelation function having a finite sum or integral.

Previous work finds, though, that wind speeds seem to exhibit "long term persistence," with autocorrelation of a hyperbolic form $\tau^{-\alpha}$ which cannot be summed. Wind's persistence was first noted by Haslett and Raftery (1989) in their analysis of

18-year wind speed records from 12 stations in Ireland. Subsequently Bakker and van den Hurk (2012) used annual-mean geostrophic winds reconstructed from 12 sea-level pressure records with 75–126 year lengths, and found statistically significant persistence over the North Sea, the British Isles and along the Scandinavian coast. Similarly, Tsekouras and Koutsoyiannis (2014) analyzed 20 observational wind-speed records from stations in Western Europe, with record lengths ranging from 65 to 107 years, and found persistence in all. These findings place year-to-year correlations in wind-speed records in the same class as

those exhibited by river flows (Hurst, 1951) and a wide variety of other climatological (Pelletier and Turcotte, 1997; Koscielny-Bunde, et al., 1998; Ault et al., 2013) and geophysical (Witt and Malamud, 2013) processes. Our findings, that interannual wind speed variability ($\hat{\sigma}/\hat{\mu}$) continues to increase with segment length out to the 62-year limit of our chronological data while for randomly permuted data it remains essentially constant, are also consistent with persistence. Because of the scale-free nature of the decay of persistent correlations, the effective number of independent samples per unit record length does not approach

a constant, and attempts to use this approach will "underrate uncertainty by a factor which tends to infinity with increasing number of observations" (Beran, 1989).

In light of the persistence behavior, it is premature to dismiss the larger estimation errors from the 60-station set as being somehow a spurious result attributable to nonstationarity. Persistent processes are characterized by seeming "trends" that spontaneously appear and disappear in a way that is actually entirely random (Beran et al., 2003, p. 3), making it difficult

to identify statistically-significant "real" trends (Cohn and Lins, 2005). Will the "stilling" trends in ground-level wind speed observations (Klink, 2002; McVicar et al., 2008; Pryor et al., 2009; Vautard et al., 2010) reverse, in the way that "global dimming" has now become "global brightening" (Müller et al., 2014)? The increasing availability of multi-decade wind speed data sets, including the Twentieth Century Reanalysis Project (Compo, 2011) in addition to the other multi-decadal works cited above, should enable an improved appraisal of the impact of trends on wind development risks.



With the rather primitive estimators utilized here, longer records do not necessarily provide a better estimate of future energy production. Since ignoring available data would not—and must not—be a reasonable solution, statistical approaches that explicitly account for the observed year-to-year correlation should be considered. One parsimonious approach would be to utilize estimation procedures based on long-term-persistence phenomenology, such as the pioneering comprehensive

5 MCP technique proposed some time ago by Haslett and Raftery (1989). Given the distinctive nature of year-to-year change in the wind resource we further suggest that it might be productive to dissociate "variability" from instrumental and model "uncertainty" in resource assessment presentations.

## 5 Data availability

The data used in this work are freely available on the Environment Canada website (Environment Canada, 2016)

10 *Acknowledgements.* We thank Environment Canada for making the monthly wind speed data used in this work openly accessible. N.B. was partially supported by a grant from Opera Universitaria of Trento. This material is based upon work funded by the National Science Foundation under Grant IIP-1332147. The authors appreciate helpful discussions with Jason Fields of the National Renewable Energy Laboratory and with Chris Gifford of DBRS Limited, Toronto.



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



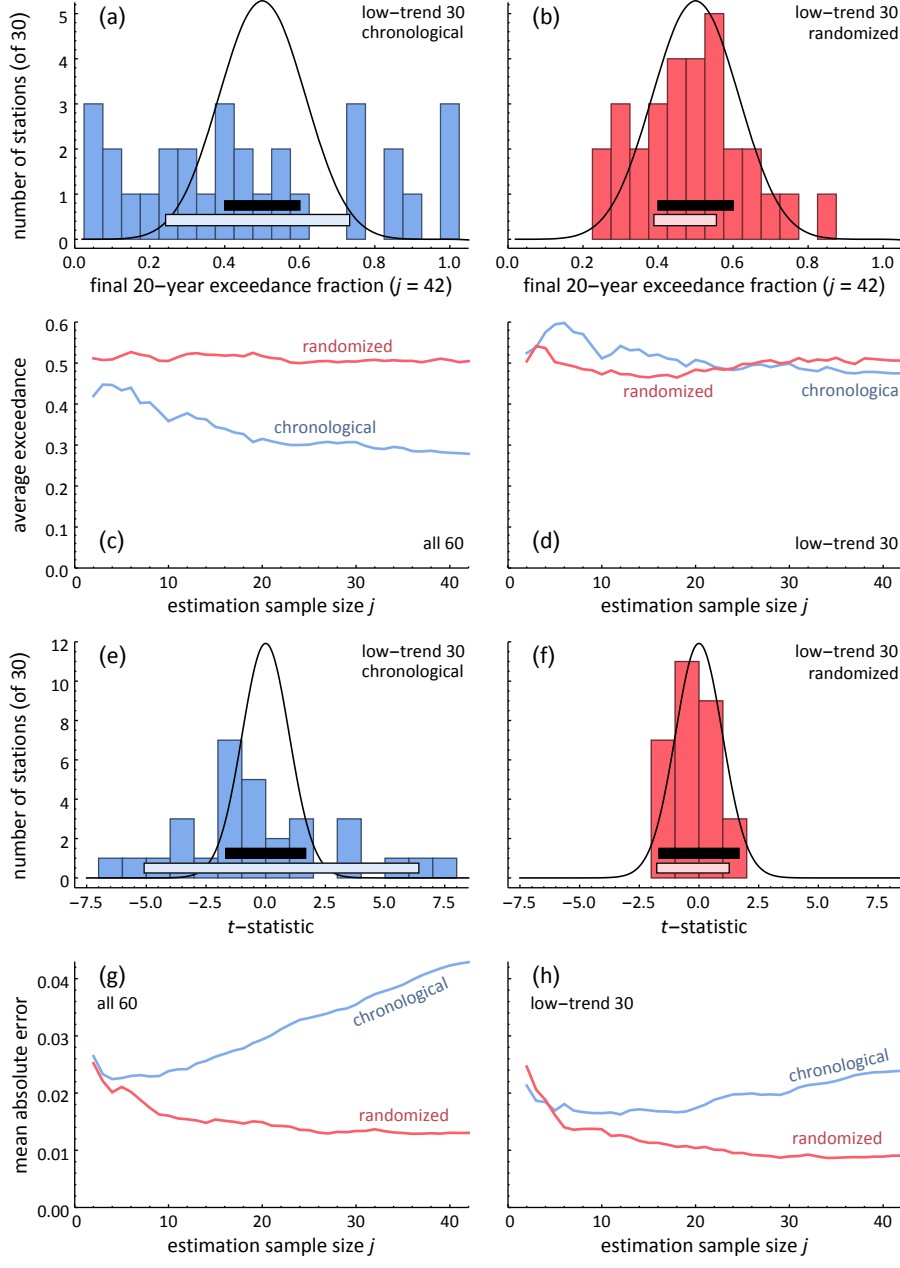

**Figure 6.** P50 estimator performance. (a), (b) Distribution of fractions of each of the 30 low-trend station's final 20 years with c.f. exceeding estimated P50 ($\hat{\mu}_{42,\ell}$) for actual and randomized data, with expected binomial distribution (curve); horizontal bars extend across the central two quartiles (binomial: black bar; data: light bars). (c), (d) P50 bias for all 60 stations, and for the 30 stations with low average trend. (e), (f) Distribution of $t$-statistic, theoretical pdf (curve), and 90% confidence intervals (theoretical: black bar; data: light bars) for difference between $\hat{\mu}_{42}$ and $\hat{M}$ for actual and randomized data, respectively, of 30 low-average-trend stations. (g), (h) P50 estimate MAE for all 60 stations and for 30 stations with low average trend.




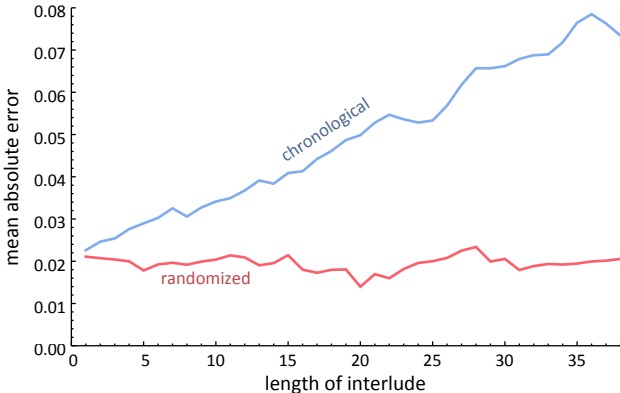

**Figure 7.** Sixty-station Mean Absolute P50 Error derived from estimate based on 5-year sample segment separated final 20-year segment by indicated interlude.

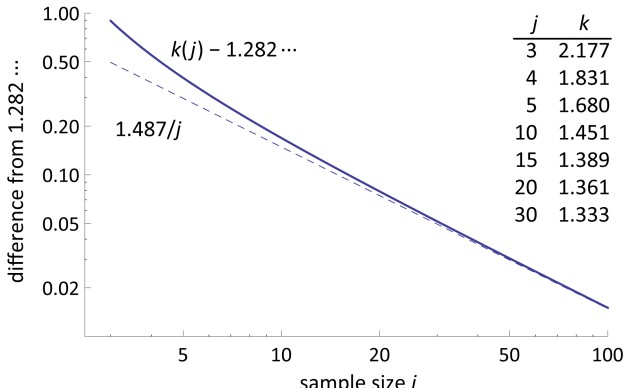

**Figure 8.** Values of $k$ (table), and difference of $k$ from 10% normal quantile vs. sample size $j$.

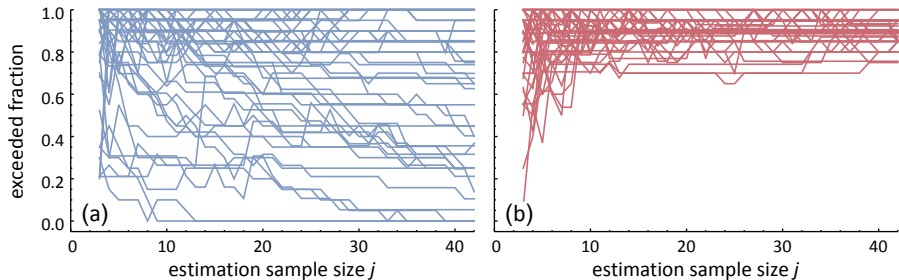

**Figure 9.** Fraction of each of the 60 stations' final 20 years with capacity factor exceeding P90 estimate for (a) actual data and (b) randomized data. The count of both years exceeding the estimate and total years were weighted according to the number of underlying data.





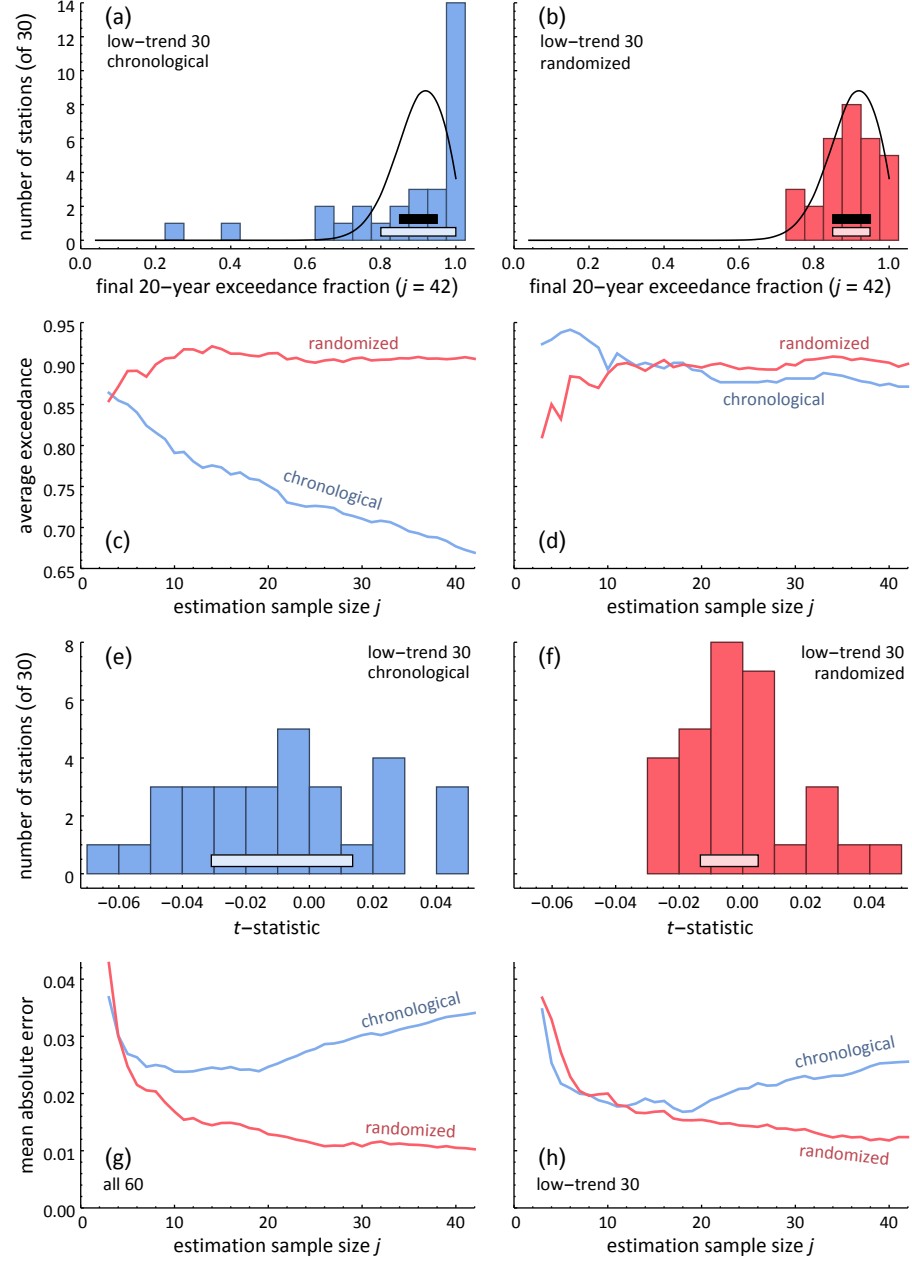

**Figure 10.** (a), (b) Distribution of fractions of each of the 30 low-trend station's final 20 years with c.f. exceeding estimated P90 ($j = 42$) for actual and randomized data with binomial PDF (curve) and central two quartiles (binomial: black bar; data: light bars). (c), (d) P90 bias for all 60 stations, and for the 30 stations with low average trend. (e), (f) Distribution of difference between $\widehat{q_{10}}$ and the dataset's 10% quantile for actual and randomized data, respectively, of 30 low-average-trend stations (light horizontal bars: central two quartiles) . (g), (h) P90 estimate MAE for all 60 stations and for 30 stations with low average trend.





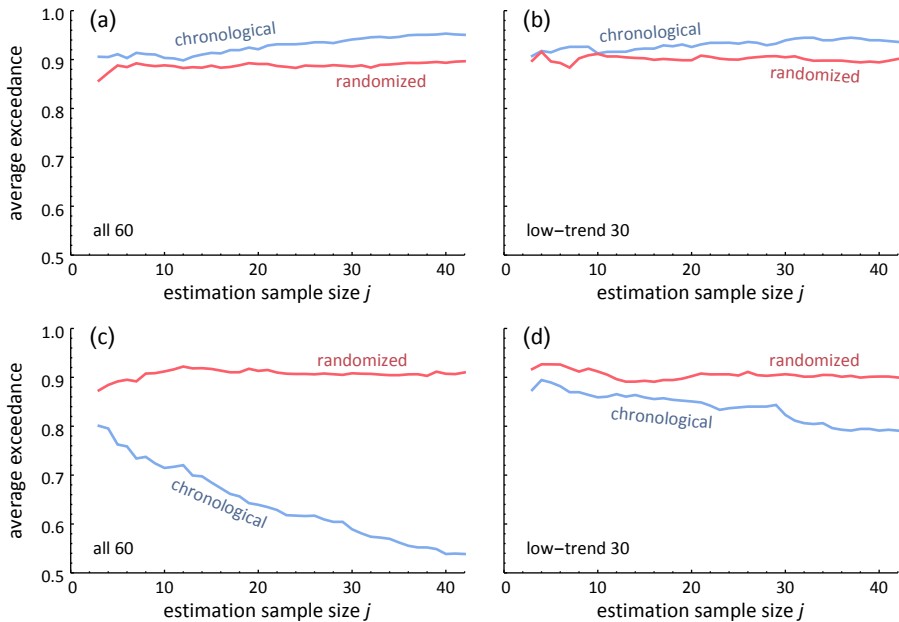

**Figure 11.** P90 "thought experiment." (a), (b) Average exceedance of P90 estimator, using the *actual mean* of the final 20 years, for all 60 stations, and for the subset of 30 stations with global near-zero trend, respectively. (c), (d) Average exceedance using the *actual standard deviation* of the final 20 years, for the 60 and 30 station sets, respectively. Traces are, but for a slight adjustment for weighting, $1/1200^{\text{th}}$ ($1/600^{\text{th}}$ for the 30 stations subset) of the count, totaled across all 60 (30) stations, of final-segment years with capacity factor exceeding each station's P90 estimate. Blue: actual data; red: randomly permuted data.