# Peer review of "Year-to-year correlation, record length, and overconfidence in wind resource assessment"

_Wind Energy Science, 2016_

## Referee Comment (RC1) · Anonymous Referee #1 · 30 May 2016

There are minor uncertainties regarding the description of the stations selection process.

In particular, when Figs 1 and 2 are compared, it seems that there are a number of stations which exhibit a capacity factor of over 0.4 during 62 years, but have been excluded due to having a capacity factor of 0.2 or less during the last 20 years. In addition, four of them also show very low variability of the order of 0.1. Is the long-term stilling trend really so strong at those stations, and are there any indication why?

Second, it is stated in P3L55 that "the cut-off also reduced the trend observed in Fig. 1 (R2=0.11)". The value of 0.11 indicates large scatter, but in the figure the filled markers seem rather organized.

And lastly, it might be better to replace circles in Fig. 3 with squares or similar, to clearly

distinguish the filling criterium from the one used in Fig. 1. There the filling corresponds to a threshold in the capacity, while here the threshold refers to the trend.

Thank you!
* * *

---

## Referee Comment (RC2) · Anonymous Referee #2 · 14 Jun 2016

This paper used a 62-year wind dataset from Canada to investigate the ability to predict P50 and P90 wind speed values for a 20 year period using different numbers of reference years. This is a very interesting topic and worthy of publication in WES.

The paper was well structured, and overall presents it work quite well. There are a few questions / comments I have that I think would help the reader better understand the work and provide a clearer presentation of some of the ideas.

1. The first time reading the abstract I had a bit of a hard time understanding the results and how the ranges should be interpreted. After reading the paper this became more clear, but I think it would help the article to clean up the abstract a bit. For example, the line "Errors of estimates made from the control sequences always decline with record length; the central half of the stations' exceedances falls within ranges of 44–55 %

(P50) and 85–95 % (P90) for 42-year estimates." is just stating that your method of predicting P50 and P90 works when using the control sequences, which perhaps does not need to be explained in such detail in the abstract.

Pg. 1 Line 23: You mention you sidestep the instrumental and model factors. It was unclear to me how this was done at this point in the paper. Later it is clear this was done using Wan et al.'s dataset, which homogenized the data, but I think it would be good to mention this earlier.

Pg. 2 Line 63: I think you should mention that they are monthly averaged wind speeds here.

Pg. 3 Lines 28-35: You mention using annual averages to avoid seasonal effects, but you then null missing months. There should be a discussion about the distribution of missing months across the year. In Canada, one could imagine that there are more missing data in the winter than the summer due to the climate, but how this might impact the data is significant. Additionally, information about how the missing years relate to the validation 20-year period compared to the fitting 42-year period may also be of interest to the reader.

Pg. 5 Lines 80-82 & figures 6 & 10: The Binomial distribution is a discrete distribution, yet you mention that the exceedances take values outside of that distribution due to your weighting and plot smooth curves on the plots mentioned above. While this shouldn't influence your findings you should correct it to properly represent the distribution.

Pg 8 Lines 8-11: This seems to be one of the most significant conclusions of your report, yet is not stated as clearly in section 4 or in the abstract. I would recommend adding it in at least one of those places as it is quite important for the resource assessment community.

---

## Referee Comment (RC3) · Anonymous Referee #3 · 17 Jun 2016

The manuscript "Year-to-year correlation, record length, and overconfidence in wind resource assessment" analyzes a 62 year, monthly averaged, surface wind speed record from Canadian stations and by comparing the original annual averaged time series to a randomized control sequence, the authors show that trends in time series can result in a substantial overconfidence in estimated resource levels.

In the title, abstract, and throughout the whole manuscript "record length" has been used. In my opinion, it would be more appropriate to replace this with "segment length", since in all experiments long term time series (62 or 42 years) have been used, splitted into different segment lengths.
P.7 l.16 Please state that the "annual averaged" wind speed data $u_i$ have been used.

P.8 l.14 It would be helpful to state explicitly that the first 42 year data has been used and not the first 42 $j$'s of the 62 year sample.

P.8 l.11 Wouldn't it be more appropriate to use the P50 instead of the mean value, since the distribution of the annual wind speeds of the original data is not normally distributed. From Fig. 6c is seems that the mean wind speed in the 42 year chronological data set increases with increasing $j$, but the P50 value would not necessarily increase.

P.9 l.9 Isn't the data sample of the first at least 5 $j$'s (number of values inside a distribution) too small to make second moment statistics?

---

## Referee Comment (RC4) · Anonymous Referee #4 · 24 Jun 2016

Review of: Year-to-year correlation, record length, and overconfidence in wind resource assessment

Climate variability is well-known in the atmospheric science community (and I think the wind energy community) so this manuscript does not report a new finding or result, but rather illustrates it (i.e. the presence of internal climate variability) again. Thus, the manuscript lacks a transformative (or novel) aspect (i.e. it illustrates a well-known challenge to the industry), it uses standard statistics (though presented in a somewhat jargon-istic manner), the results are not generalizable (i.e. the uncertainties in P50 and P90 are inevitability going to be location specific) and it does not present a 'path-forward' for the industry or advance diagnostic understanding of causes of uncertainty in P50 and P90. Further, use of monthly mean wind speeds at 10-m (presumably also confounded by instrumentation and other changes) renders the analysis 'results'

very highly suspect for real-world applications. For these reasons I do not recommend publication in a high-profile international journal.

---

## Author Comment (AC1) · 30 Jun 2016

**Response to Referee 3**

**RC3:** In the title, abstract, and throughout the whole manuscript "record length" has been used. In my opinion, it would be more appropriate to replace this with "segment length", since in all experiments long term time series (62 or 42 years) have been used, splitted into different segment lengths.

**Authors:** We actually distinguish between "record length" and "segment length," and use the two terms to indicate different things in our paper. We already use "segment length," as the referee suggests, in our text with regard to Figure 7, where we split

the data sequence. In other places we use the term "record length" to mean the uninterrupted sequence of years immediately preceding the prediction. Although when we refer to short "record lengths" we are clearly not using all the data, we believe this usage speaks most clearly to the important question in wind resource-assessment practice of the minimum historical record needed to achieve a desired level of certainty.

**RC3: P.7 l.16** Please state that the "annual averaged" wind speed data $u_i$ have been used.

**Authors:** We have revised the sentence immediately following Equation (4) to read,

where the $w_i$ are the normalized data-availability weights of the annual averaged wind-speed data $u_i$, and $j$ indicates the sample size (length in years of the data segment over which IAV is calculated).

**RC3: P.8 l.14** It would be helpful to state explicitly that the first 42 year data has been used and not the first 42 $j$'s of the 62 year sample.

**Authors:** We are not sure we understand the referee's suggestion, but have revised the text in the hope of making our meaning more explicit. It now reads :

To evaluate how P50 estimation performance depends on record length, we calculated, for each selected station ($\ell = 1, \ldots, 60$ or, for the low-trend subset, $\ell = 1, \ldots, 30$), the weighted mean $\hat{\mu}_{j,l}$ of the immediately preceding $j$ annual capacity factors ($j = 1, \ldots, 42$), and then counted the (weighted) number of the final 20 years having a capacity factor exceeding $\hat{\mu}_{j,l}$. For example, $\hat{\mu}_{5,11}$ signifies a P50 estimate predicting years 43 to 62 of station 11's performance, the estimate calculated as the resource-level average

over years 38–42.

**RC3: P.8 I.11** Wouldn't it be more appropriate to use the P50 instead of the mean value, since the distribution of the annual wind speeds of the original data is not normally distributed. From Fig. 6c is seems that the mean wind speed in the 42 year chronological data set increases with increasing j, but the P50 value would not necessarily increase.

**Authors:** If non-normality (or more essentially, non-zero skew) of the distribution of original annual wind speeds were significant, it would create systematic bias in the red trace in Figure 6(c). The data used to calculate the red trace have exactly the same 62 values for each station as the data for the blue trace; only their order for a given station is different–thus the distributions are exactly the same. The fact that the red trace exhibits no significant bias or trend (i.e. that it varies slightly but randomly around the intended value of 50%) means that no such skew is present. Therefore, the trend in the blue trace could arise only from the ordering of the data, not from its distribution. Thus, a P50 prediction calculated from the historical median rather than the historical mean would exhibit the same trend.

**RC3: P.9 I.9:** Isn't the data sample of the first at least 5 $j$'s (number of values inside a distribution) too small to make second moment statistics?

**Authors:** The text on p. 9 near line 9 refers to bias in the P50 estimate. The P50 estimate is a first-moment statistic (sample mean). Its bias is calculated by averaging P50 error from each of 60 stations. If we correctly understand what the referee is referring to, we believe 5 year's data from 60 stations is sufficient to evaluate bias.

---

## Author Comment (AC2) · 30 Jun 2016

**Response to Referee 4**

**RC4:** ... this manuscript does not report a new finding or result, but rather illustrates it (i.e. the presence of internal climate variability) again. Thus, the manuscript lacks a transformative (or novel) aspect (i.e. it illustrates a well-known challenge to the industry), it uses standard statistics (though presented in a somewhat jargon-istic manner), the results are not generalizable (i.e. the uncertainties in P50 and P90 are inevitability going to be location specific) and it does not present a 'pathforward' for the industry or advance diagnostic understanding of causes of uncertainty in P50 and P90. Further, use of monthly mean wind speeds at 10-m (presumably also confounded by

instrumentation and other changes) renders the analysis 'results' very highly suspect for real-world applications.

**Authors:** We respectively disagree with the opinion of Referee 4. The challenge to the wind industry is to translate climate variability (the existence of which we agree needs no further illustration) into quantitative assessments of financial risk. It is not the purpose of our paper to survey the troubled history of resource assessment, but a glance at the following recent reports from the leading engineering firms and financial institutions evidences the industry's continuing concern over widespread risk- and performance-assessment errors. It is far from "old hat."

1. C. J. Kim, "Breezing Past P50," Moody's Investor Service (2010). *Forecasts for US based wind projects appear to have been overly optimistic since actual production is far below the expected level of generation typically represented by the P50.*

2. N. G. Mortensen, H. E. Jørgensen, M. Anderson, and K.-A. Hutton, "Comparison of Resource and Energy Yield Assessment Procedures," presented at the EWEA Annual Conference and Exhibition, Copenhagen, 2012, http://orbit.dtu.dk/ws/files/10376800/Comparison_of_Resource.pdf.

3. AWS Truepower, "Closing the gap on plant underperformance," (revs'd 2012). *The study found that the methods in place at AWS Truepower at that time, though much improved over prior methods, overestimated actual energy production by an average of 3.5%, after correction for relative windiness.*

4. N. G. Mortensen, and H. E. Jørgensen, "Comparative Resource and Energy Yield Assessment Procedures (CREYAP) Pt. II," presented at the EWEA Technology Workshop: Resource Assessment 2013, Dublin, 2013, http://orbit.dtu.dk/files/70667004/Comparative_Resource_and_Energy_Yield.pdf.

5. J. Babajeva, G. Remec, F. Gronda, C. Kuti, and N. Czarny, "Wind Projects: High Risk of Production Shortfalls," FitchRatings, https://www.fitchratings.com/creditdesk/reports/report_frame.cfm?rpt_id=749633 (2014). *Fitch Ratings compared the performance of 19 operating wind projects with its expectations when the ratings were assigned. There are chronic production shortfalls below base case expectations at a majority of the projects. Actual production only occasionally exceeds base case levels. . . .*

6. R. Z. Poore, "Wind Power Project Performance White Paper: Actual vs. Predicted," DNV GL - Energy, Renwables Advisory, http://www.gl-garradhassan.com/assets/img/content/DNV_GL_Wind_Power_Project_Performance_White_Paper.pdf (2014).

7. R. Istchenko, "Yield and Uncertainty Validation for Pre- and Post Construction Wind Resource Assessment," presented at the AWEA WindPower 2014, Las Vegas, http://www.phoenixengg.com/Publications/2014_YieldAndUncertaintyValidation.pdf 2014. *Historically, project yields haven't met expectations: bias of 8-10%.*

8. M. Stoelinga, and M. Hendrickson, "A Validation Study of Vaisala's Wind Energy Assessment Methods," Vaisala, http://www.vaisala.com/en/press/news/2015/Pages/Page_1973472.aspx (2015). *For the last decade, underperformance has been a key concern of the wind industry and investigations have revealed that more sophisticated assessment methods are required to improve pre-construction energy estimates during the due diligence phase.*

9. A. Clifton, A. Smith, and M. Fields, "Wind Plant Preconstruction Energy Estimates. Current Practice and Opportunities," National Renewable Energy Laboratory, NREL/TP–5000-64735 (2016).

The novel aspect of our paper is that it shows for the first time that year-to-year

correlations in wind resource level have a *large* effect on resource assessment uncertainty. The presence of these correlations is largely ignored in current industry practice. Our identification of this effect constitutes a very fundamental advance in the diagnostic understanding of causes of uncertainty in P50 and P90. This finding is also very general: the identified correlations were present at all 60 sites in the present study, and are in our opinion highly likely to present at every wind site in the world. Recognizing the existence of such correlations links wind resource assessment to a wide body of statistical understanding of related natural phenomena in other fields, and thus indeed provides a 'pathforward,' as we suggest with regard to references we cite, namely Hurst (1951), Beran (1989, 2003), Haslett and Raftery (1989), Pelletier and Turcotte (1997), Koscielny-Bunde, et al. (1998), Ault (2013), Witt and Malamud (2013), and Tsekouras and Koutsoyiannis (2014).

The concern over the underlying data being "presumably also confounded by instrumentation and other changes" can best be alleviated by referring to the paper accompanying the source data (Wan, 2010), which describes the state-of-the art homogenization techniques employed.

---

## Author Comment (AC3) · 30 Jun 2016

Response to Referee 2

We thank the referee insightful comments following a thorough reading of our ms.

RC2: I think it would help the article to clean up the abstract a bit...

Authors: We agree with all the reviewer's points and have substantially revised the abstract. Our revised abstract now reads as follows:

Interannual variability of wind speeds presents a fundamental source of uncertainty in preconstruction energy estimates. Our analysis of a set of long (62-year) wind-speed records from 60 stations in Canada shows that deviations from mean resource levels persist over many decades and make actual uncertainty 2-3 times larger than

[Figure]

expected. For example, the performance of each site's last 20 years diverges widely from the P50 level estimated from its first 42 years: half the sites have either fewer than 5 or more than 15 years exceeding the P50 estimate. In contrast to this 10-year-wide interquartile range, a 4-year-wide range (2.5 times narrower) was found for "control" records where statistical independence was enforced by randomly permuting each station's historical values. Similarly, for sites with capacity factor of 0.35 and interannual variability of 6%, one would expect 9 years in 10 to fall in the range 0.32-0.38; we find the actual 90% range to be 0.27-0.43, or three times wider. The presence of serial correlations favors a shift from a climatology-focused approach to a persistence-focused approach: for this dataset, no improvement in P50 error is gained by using records longer than 4-5 years, and use of records longer than 20 years actually degrades accuracy.

RC2: Pg. 1 Line 23: You mention you sidestep the instrumental and model factors. It was unclear to me how this was done at this point in the paper. Later it is clear this was done using Wan et al.'s dataset, which homogenized the data, but I think it would be good to mention this earlier.

Authors: The main point we were trying to make here actually does not refer to Wan's homogenization of the data. We have revised this paragraph to indicate that, by predicting simply just the same variable that has been measured (anemometer wind speed), we avoid the additional uncertainties associated with modeling flow over complex terrain, turbine availability, and so on, that complicate full resource assessments but that do not relate to wind's predictability. The revised text reads as follows:

To avoid uncertainties associated with the full energy assessment, such the modeling of flow over complex terrain, extrapolation of wind speeds to hub height, wake losses, turbine availability, and so on, we analyze only the extent to which historical anemometer measurements can be used to predict future ranges of the same anemometer measurements. (We convert anemometer measurements to modeled turbine capacity factor just to give an appropriate variability scale.)

RC2: Pg. 2 Line 63: I think you should mention that they are monthly averaged wind speeds here.

Authors: we have inserted the word "average". The text now reads, "We base our analysis here on one of the longer observational datasets of instrumental wind speed records available, a 62-year (1953–2014) record of monthly-average wind speeds from 156 Canadian meteorological stations (Environment Canada, 2016)."

RC2: Pg. 3 Lines 28-35: You mention using annual averages to avoid seasonal effects, but you then null missing months. There should be a discussion about the distribution of missing months across the year. In Canada, one could imagine that there are more missing data in the winter than the summer due to the climate, but how this might impact the data is significant. Additionally, information about how the missing years relate to the validation 20-year period compared to the fitting 42-year period may also be of interest to the reader.

Authors: We appreciate the reviewer's point about the significance of averaging over possibly non-uniform distributions of missing months. We have accordingly revised the method by which calculate a station's annual resource-level average for years in which one or more monthly data values are missing. To describe this revised method, we have added after the original sentence, "To deal with data gaps, unavoidably present in such an extended dataset, we assign null weights to missing monthly data," the following new text:

Furthermore, for each station we calculate an average seasonal cycle as the average of all the Januarys, all the Februarys, and so on (for both wind speed and modeled capacity factor) over the station's 62-year record. In years where our annual average is calculated from less than 12 data points, we make an adjustment according to the fraction the available data represent of the average seasonal cycle.

RC2: Pg. 5 Lines 80-82 & figures 6 & 10: The Binomial distribution is a discrete distribution, yet you mention that the exceedances take values outside of that distribution

due to your weighting and plot smooth curves on the plots mentioned above. While this shouldn't influence your findings you should correct it to properly represent the distribution.

Authors: We have revised figures 6 & 10, as shown, to more explicitly show the discrete nature of the binomial distribution.

[Figure]

[Figure]

[Figure]

**Fig. 1.**

[Figure]

[Figure]

**Fig. 2.**

[Figure]

---

## Author Comment (AC4) · 30 Jun 2016

Response to Referee 1

We thank the referee for comments that have helped us clarify our presentation, as we indicate below:

RC1: . . . it seems that there are a number of stations which exhibit a capacity factor of over 0.4 during 62 years, but have been excluded due to having a capacity factor of 0.2 or less during the last 20 years. In addition, four of them also show very low variability of the order of 0.1. Is the long-term stilling trend really so strong at those stations, and are there any indication why?

Authors: In fact, these high-capacity-factor stations you note were excluded not because of capacity factor being lower than 0.2 during the last 20 years, but because of poor data quality, as described in the ms.: "We further eliminate another 13 stations that have more than a single isolated year with no or scant monthly data." (Pg. 4, Line 22).

RC1: Second, it is stated in P3L55 that "the cut-off also reduced the trend observed in Fig. 1 ($R^2$ = 0.11)". The value of 0.11 indicates large scatter, but in the figure the filled markers seem rather organized.

Authors: We have corrected the value in the ms. to $R^2$ = 0.15.

RC1: And lastly, it might be better to replace circles in Fig. 3 with squares or similar, to clearly distinguish the filling criterium from the one used in Fig. 1. There the filling corresponds to a threshold in the capacity, while here the threshold refers to the trend.

Authors: The symbol filling scheme we chose here helps the viewer easily visualize the marginal distributions of the selected low-trend stations (filled circles), while minimizing distraction from the excluded stations (unfilled symbols), so we're reluctant to change the plotting scheme. But, we have added legends, as shown in the accompanying figures, to ensure there's no confusion.

―――――――――――――――――――

**Fig. 1.** Figure 1, new version

Plot area:
- Y-axis: $\Delta\hat{\mu}$ with values 0.05, 0.00, −0.05, −0.10
- X-axis: 62−year average capacity factor $\mu$ with values 0.20, 0.25, 0.30, 0.35, 0.40, 0.45, 0.50

Legend:
- ● low−trend subset
- ○ other stations

**Fig. 2.** Figure 3, new version

---

## Author Response (AR1)

We thank the Editor for his decision that our paper should be published.

**1  Point-by-point response to referees**

**1.1  RC1**

In response to points made by the first referee, we have:

- corrected to 0.15 the $R^2$ value associated with the fit in Figure 1 to the data for the selected 60 stations, and
- added legends to Figures 1 and 3.

**1.2  RC2**

In response to points made by the second referee, we have:

- substantially revised the abstract,
- added text in the Introduction to clarify how we avoid the uncertainties associated with site modeling and with energy production by simply using anemometer wind-speed records to predict expected future ranges again of anemometer wind speeds (with conversion to turbine capacity factors being used to set the correct variability scale),
- completely re-done our analyses using an improved method of calculating annual-average wind speeds and capacity factors when data for one or more months of a year are missing, as explained in a new sentence following Equation (2),
- made minor revisions to numerical claims in the paper in accordance with the above revised analyses, and
- re-drawn Figures 6 and 10 to clearly indicate the discrete nature of the Binomial distribution.

**1.3  RC3**

In response to points made by the third referee, we have:

- added a sentence in Section 3.2 providing a concrete example of our $\hat{\mu}_{j,\ell}$ notation for the P50 estimator.

**2  Response to Editor's request for minor revisions**

In response to a request from the Editor referring to remarks by Referee 4, we have:

- added text to the Abstract pointing out that one of the novel features of our work is its reliance on an especially large dataset with exceptionally long instrumental windspeed records covering a continental-sized geographic region,
- added text to the Abstract pointing out that another novel feature of our work is that the effect of serial correlation on wind resource assessments has not previously been quantified, and
- revised the concluding paragraph of the paper by adding the sentence, "We have shown here that year-to-year correlations in resource level produce large effects, seemingly not recognized or incorporated into current estimation practice, degrading the certainty of pre-construction wind energy estimates."

These changes are indicated on the marked-up version of the manuscript on the following pages.

[revised manuscript text omitted]